# Bridging the Gap: An Algorithmic Framework for Vehicular Crowdsensing

**DOI:** 10.3390/s24227191

**Published:** 2024-11-09

**Authors:** Luis G. Jaimes, Craig White, Paniz Abedin

**Affiliations:** Department of Computer Science, Florida Polytechnic University, Lakeland, FL 33805, USA; cwhite@floridapoly.edu (C.W.); pabedin@floridapoly.edu (P.A.)

**Keywords:** vehicularcrowdsensing, greedy algorithms, recurrent reverse auctions

## Abstract

In this paper, we investigate whether greedy algorithms, traditionally used for pedestrian-based crowdsensing, remain effective in the context of vehicular crowdsensing (VCS). Vehicular crowdsensing leverages vehicles equipped with sensors to gather and transmit data to address several urban challenges. Despite its potential, VCS faces issues with user engagement due to inadequate incentives and privacy concerns. In this paper, we use a dynamic incentive mechanism based on a recurrent reverse auction model, incorporating vehicular mobility patterns and realistic urban scenarios using the Simulation of Urban Mobility (SUMO) traffic simulator and OpenStreetMap (OSM). By selecting a representative subset of vehicles based on their locations within a fixed budget, our mechanism aims to improve coverage and reduce data redundancy. We evaluate the applicability of successful participatory sensing approaches designed for pedestrian data and demonstrate their limitations when applied to VCS. This research provides insights into adapting greedy algorithms for the particular dynamics of vehicular crowdsensing.

## 1. Introduction

Mobile crowdsensing systems traditionally harness the sensing capabilities of cellular phone users to collect and transmit data on various environmental parameters to a central monitoring entity. These systems leverage the extensive user base to gather significant volumes of data from previously inaccessible locations, thereby addressing large-scale societal issues [1,2,3]. A common application involves using pedestrian crowdsensing to monitor air quality, thereby assessing pollution levels across different geographical scales, from local neighborhoods to entire countries.

With the advent of smart mobility (connected, autonomous, and semi-autonomous), there is growing interest in transitioning from pedestrian to vehicular crowdsensing. This shift capitalizes on the increasing prevalence of vehicles equipped with advanced sensing technologies, such as GPS and environmental sensors. Vehicular crowdsensing can provide more comprehensive spatial coverage and real-time data collection compared to pedestrian systems. For instance, vehicles can continuously monitor air quality over broader areas and varying traffic conditions, offering valuable insights into pollution patterns and traffic-related environmental impacts.

A primary challenge in both pedestrian and vehicular crowdsensing systems is ensuring active participant engagement. In these systems, users typically do not receive direct benefits from reporting data, which can impact their willingness to participate [4,5]. Effective incentive mechanisms are thus crucial to sustaining participation.

Reverse auctions are a prevalent method for incentivizing participants, including in vehicular crowdsensing scenarios [6]. This approach involves participants competing to offer their sensing data, with the auctioneer purchasing the cheapest *m* samples. This method helps manage costs but can lead to participant attrition if not properly managed. The development of incentive mechanisms is required to mitigate this risk and ensure system reliability.

Current incentive schemes, while effective in some contexts, often overlook critical factors such as the spatial distribution of data sources, coverage, and budget constraints. For example, purchasing the *m* least expensive samples in each auction round might lead to acquiring data from geographically clustered sources, resulting in redundant information. To address this, incentive mechanisms should incorporate location and coverage constraints to ensure that data are not only cost-effective but also spatially representative. Additionally, budgetary constraints must be considered, as assuming an unlimited budget is impractical.

In this paper, we investigate the application of incentive mechanisms originally designed for mobile crowdsensing with pedestrian participants within the domain of vehicular crowdsensing (VCS). Utilizing open-street maps, the Simulation of Urban Mobility (SUMO) vehicular traffic simulator, and comprehensive simulations, we demonstrate that results and insights derived from pedestrian-based crowdsensing do not directly apply to the vehicular crowdsensing context. As a general framework for data purchasing, we use a recurrent reverse auction [6]. For acquisition optimization or recruitment, we use four different greedy strategies [7] and a random one. Finally, we show that the positive results of the strategy that overwhelmingly outperformed in the context of pedestrians did not translate in the context of vehicular data collection.

The remainder of this paper is organized as follows: Section 2 provides a brief review of the relevant literature. Section 3 introduces the system model. Section 4 details the performance evaluation within the context of pedestrian-based crowdsensing. Section 4.2 extends this evaluation to the context of vehicular crowdsensing. Finally, Section 5 summarizes the findings and offers recommendations for future research.

## 2. Literature Review

This section provides a brief overview of prior work and advancements in crowdsensing.

With the rapid advancements in Internet of Things (IoT) technology, a wide range of mobile devices such as smartphones, vehicle terminals, and drones are now equipped with sophisticated sensors and wireless communication modules, including microphones, GPS, cameras, and gyroscopes. This technological progress has given rise to a novel data-sensing approach known as mobile crowdsensing (MCS). MCS leverages mobile devices as primary sensing units, distributing sensing tasks and collecting data through the Internet. This approach facilitates the completion of various complex and large-scale sensing and computing tasks [8,9].

In “Sensing Data Market”, Chou, Bulusu, and Kanhere [10] highlight the need for incentive mechanisms to motivate individuals to share their sensing data. Additionally, they foresee the implementation of mechanisms to ensure data quality, preventing redundancy and coverage issues. However, the authors only identify these requirements as essential for establishing a sensing data market, without detailing any specific mechanisms for their implementation.

Other authors [11,12] employed economic and psychological methods to tackle the issue of incentivizing individuals to share and utilize their location data. Participants in their experiment were made to think that a sealed-bid second-price auction was being conducted to ascertain the price at which they would reveal their location (privacy). The research identified the rates of non-participation and dropout, which are crucial for understanding the value users place on their location data. This study serves as a reference for establishing the parameters of the reverse auction scheme proposed here.

Authors like Shaw et al. [13] and Yan et al. [14] examine crowdsourcing as a method for recruitment, while Reddy et al. [15] investigate the use of micro-payments—transactions where small tasks are paired with small payments, as a form of incentive. In this study, a pilot experiment was conducted with 55 participants, utilizing 5 different types of micro-payments: a lump sum payment (MACRO), medium micro-payment (MEDIUM), high micro-payment (HIGH), low micro-payment (LOW), and competition-based (COMPETE), with the latter being the only dynamic method. The experiment aimed to gain insights into recycling practices.

Participants were instructed to take photos of the contents of outdoor waste bins and label the images with tags indicating their contents. The MACRO scheme promised individuals $50 to participate in the study. The MEDIUM, HIGH, and LOW schemes offered 20, 50, and 5 cents per valid submission, respectively. The COMPETE scheme’s payment was based on peer ranking, determined by the number of samples taken, and ranged from 1 to 22 cents per valid submission. The total payout for the micro-payments was capped at $50 per participant. The results showed that the HIGH and MEDIUM schemes were the most successful, while the MACRO and LOW schemes resulted in fewer photo submissions, and several participants in the COMPETE group dropped out. This study indicates that flat rates, like the MACRO scheme, do not encourage user participation, and dynamic schemes like COMPETE lead to fatigue and participant dropout.

Experiments conducted by Lee and Hoh [6] highlight the challenge of determining the appropriate amount for fixed pricing (micro-payments), as a high price can render the strategy economically unviable, while a low price can discourage user participation. They also found that dynamic schemes tend to lose participants over time. To address these issues, they propose the Reverse Auction-Based Dynamic Price with Virtual Participation Credit and Recruitment scheme (RADP-VPC-RC), which allows individuals to sell their sensing data through a dynamic price reverse auction system. This scheme incorporates several mechanisms to maintain a minimum number of participants, accommodate people with different reward expectations, and recruit members who have dropped out, addressing common issues in recurrent reverse auction systems. Although RADP-VPC-RC reduces sample prices through participant competition, it does not tackle problems of data redundancy, coverage, and budget constraints, which are the focus of our study. The RADP-VPC-RC scheme is further detailed in the next section, as our cost-effective Greedy Incentive Algorithm (GIA) [16] scheme is based on it.

Yang et al. [17] introduced the concept of platform-based mobile crowdsensing. In this work, the authors model a crowdsensing system as a Stackelberg game, a sequential game where players are divided into leaders and followers. The game is structured as a two-level model, with the platform as the leader (first player) and the vehicles as the followers. The second level of the game corresponds to a non-cooperative scenario where the participants (players) compete against each other to maximize their profits. However, Yang’s work considers only one platform and multiple players. Chakeri et al. [18,19] extend Yang et al.’s work by developing a greedy algorithm that enables the implementation of a crowdsensing system using multiple platforms and participants per platform. This foundational work would later become crucial for advancing vehicular crowdsensing systems.

Building upon the work of Chakeri et al. [18], another branch of vehicular crowdsensing research utilizes Nash Equilibrium (NE), a key concept in game theory. In this approach, the crowdsensing system is modeled as a game, where players assume two distinct roles: participant vehicles and the platform. The platform’s role involves designating areas of sensing interest and assigning corresponding rewards, while the participants are responsible for data collection. Representative work in this field includes studies such as [5,20,21,22]. Unlike the work of Chakeri and Yang, where participants are static and the games are one-shot, this new body of work focuses on evolutionary games. In these games, vehicular players are dynamic, requiring them to adjust their strategies as the game progresses.

One of the other main research challenges in mobile crowdsensing (MCS) involves ensuring sufficient coverage of sensing tasks, which reflects the effectiveness of the incentive mechanism designed to maximize this coverage [5]. Another key challenge is determining the optimal number of mobile users required to complete a given task within a defined area of interest [23]. Several studies have been conducted to improve coverage quality through the implementation of budget-constrained optimal participant selection strategies.The authors Zheng et al. [24] and Zhang et al. [25] proposed a greedy method to recruit an optimal set of mobile users who can cover the entire region of interest while keeping within the system’s budget constraints. Meanwhile, Xu et al. [26] presented budget-feasible frameworks for selecting an ideal group of mobile users to maximize continuous-time interval coverage while adhering to budgetary limits. In this paper, we address the dual challenges of ensuring sufficient coverage of sensing tasks and effectively incentivizing participants in mobile crowdsensing (MCS). We focus on developing strategies that not only maximize the coverage of sensing tasks but also create robust incentive mechanisms to engage and retain participants.

## 3. System Model

We model our VCS system as a game with two main players: a crowdsourcer or data buyer $C_{i}$, and a set of participant vehicles $V=\{v_{1},v_{2},\ldots ,v_{N}\}$. The crowdsourcer aims to collect a representative set of samples $S=\{s_{1},s_{1},\ldots ,s_{M}\}$ with M≤N of a given region $G_{i}$ to reconstruct a variable of interest. For that purpose, the crowdsourcer uses a limited budget *L* to outsource the data collection to a set of interested participants *V*. Given the limited budget available, the crowdsourcer cannot purchase all the samples offered by participants at any given time, and its goal is to acquire a sub-collection of samples $S=\{S_{1},S_{2},\ldots ,S_{M}\}$ that covers as many of the available samples as possible, resulting in the acquisition of a representative set of samples of the area of interest within a given budget. For that purpose, we use a model (GIA) with three main components: a geometric sensor model, a multi-round reverse auction method for data buying or acquisition, and several recruitment approaches for selecting the winners of each round.

### 3.1. Geometrical Sensor Model

To address the coverage problem, we utilize the following geometric disk model:(1)$$f\left(d\left(u_{i},u_{j}\right)\right)=\left\{\begin{array}{cc} 1 & \mathrm{if}\;d(u_{i},u_{j})\leq r\\ 0 & \mathrm{otherwise} \end{array}\right.$$

In this model, $d(u_{i},u_{j})$ denotes the Euclidean distance between sensor $u_{i}$ and sensor $u_{j}$, and r>0 is a constant that defines the coverage radius for each sensor. This function characterizes a disk centered at sensor $u_{i}$ with radius *r*. Sensors located within this disk are assigned a coverage measure of 1 and are considered covered by sensor $u_{i}$. In contrast, sensors situated outside this disk have a coverage measure of 0 and are regarded as not covered by $u_{i}$. Figure 1 illustrates the coverage areas of sensors 1 and 7. The fundamental premise is that the sensing value of any environmental variable within a disk of radius *r* is uniform. Consequently, if sensing samples are obtained from sensors 1 and 7, it is redundant to collect additional samples from sensors 2, 3, 4, 5, and 6, as their data would be similar.

### 3.2. Data Purchasing Model

For data purchasing, we use the reverse Auction-based Dynamic Price with Virtual Participation Credit and Recruitment (RADP-VPC-RC) [6,16,27,28]. Algorithm 1 describes a round of the multi-round auction-based mechanism.
**Algorithm 1** RADP-VPC-RC incentive mechanism  1:Initialize auction round r←1  2:Initialize Virtual Participation Credit $v_{i}\leftarrow 0$ for all users *i*  3:Set number of required participants *Q*  4:**while** service quality is not guaranteed **do**  5:    **for** each user *i* **do**  6:        Submit actual bid $b_{i}^{r}$  7:        Calculate competition bid $b_{i}^{r}\leftarrow b_{i}^{r}-v_{i}$  8:    **end for**  9:    Sort users based on competition bids $b_{i}^{r}$10:    Select the top *Q* users as winners11:    **for** each winner *i* **do**12:        Pay $b_{i}^{r}$ to participant *i*13:        Reset $v_{i}\leftarrow 0$14:        increase the $b_{i}\leftarrow 1.1b_{i}$ with 50% of probability15:    **end for**16:    **for** each loser *i* **do**17:        Update Virtual Participation Credit $v_{i}\leftarrow v_{i}+\alpha$18:        decrease the $b_{i}\leftarrow 0.8b_{i}$ with 50% of probability19:        evaluates ROI if ROI≥0.5 stays, otherwise it drops20:    **end for**21:    Notify winners and losers22:    Send the highest bid price to the dropped users for recruiting23:    Dropped users evaluate their ROI, if ROI≥0.5, return with 50% of probability in next round.24:    r←r+125:**end while**

Line 1 initializes the first round, and Line 2 sets the Virtual Participation Credit $v_{i}$ to zero for all participants *i*. In Lines 4 to 8, every active participant *i* submits its bid’s price $b_{i}^{r}$ (Bid Price of Participant i at Round r) for its sample; however, its competition bid’s price corresponds to its bid’s price minus its accumulated Virtual Participation Credit. Lines 9 and 10 show how to select the winners for the current round *r* (the selection criteria could be modified according to the recruitment algorithm). Lines 10–14 describe how the Virtual Participation Credit of winners is set to zero, and half of the winners randomly increase their bid’s price by 10%. Lines 16–19 describe how the losers’ Virtual Participation Credit is increased, resulting in a more competitive bid price and more chances to win in the next round. In addition, half of the losers randomly decrease their bid’s price to improve the winning chances for the next round. Finally, Lines 20–23 describe the re-joint process. Here, the system sends the highest bid price of the current price to the dropped users, which is used for them to evaluate their Return on Investment (ROI); if it is greater than 0.5, then they randomly return to the next round with a 50% probability. Finally, Line 24 signals the transition to the next round where the whole process re-takes place. A detailed description of some elements of this reverse auction such as the ROI and others can be found in [16].

### 3.3. Recruitment Model

We compare four approaches to select each round’s winners, specifically, the participants from whom the samples are purchased. The first approach is the Greedy Budgeted Maximum Cover (GBMC) or cost-effective approach, which is the core of the GIA (Greedy Incentive Algorithm). The second approach corresponds to a pure greedy strategy that acquires the cheapest samples until all the samples are obtained or the budget is exhausted, which is the core RADP-VPC-RC. The third approach uses a GBMC with enumeration with k = 1 and k = 2. Finally, the fourth approach is a random acquisition method in which no optimization strategy is used to select the winners for each round. The next subsection briefly reviews each of the recruitment strategies.

#### 3.3.1. Greedy Budgeted Maximum Coverage Algorithm (GBMC) or the Cost-Effective Algorithm

GIA [16] uses RADP-VPC-RC for data acquisition, and the Greedy Budgeted Maximum Coverage Algorithm (GBMC) [7] for recruitment. For the reader’s ease, we present the GBMC algorithm in Algorithm 2.
**Algorithm 2** The Greedy Budgeted Maximum Coverage Algorithm for GIA.  1:**Input:** *S* a collection of sets made up by the user locations  2:**Output:** $S^{\prime }\subseteq S$, covering set  3:**function** GREEDY AREA PRICE(*S*)  4:    G←∅  5:    C←0  6:    U←S  7:    **while** U≠∅ **do**  8:        select $S_{i}\in U$ that maximizes $\frac{W_{i}^{\prime }}{c_{i}}$  9:        **if** $C+c_{i}\leq L$ **then**10:              $G\leftarrow G\cup S_{i}$11:              $C\leftarrow C+c_{i}$12:        **end if**13:        $U\leftarrow U\setminus S_{i}$14:    **end while**15:    **return** *G*16:**end function**17:**function** GREEDY AREA(*S*)18:    $G^{\prime }\leftarrow \emptyset$19:    $C^{\prime }\leftarrow 0$20:    $U^{\prime }\leftarrow S$21:    **while** $U^{\prime }\neq \emptyset$ **do**22:        select $S_{j}\in U^{\prime }$ that maximizes $W_{j}^{\prime }$23:        **if** $C^{\prime }+c_{j}^{\prime }\leq L$ **then**24:              $G^{\prime }\leftarrow G^{\prime }\cup S_{j}$25:              $C^{\prime }\leftarrow C^{\prime }+c_{j}^{\prime }$26:        **end if**27:        $U^{\prime }\leftarrow U^{\prime }\setminus S_{j}$28:    **end while**29:    **return** $G^{\prime }$30:**end function**31:**function** GBMC32:    G←GREEDY AREA PRICE(*S*)33:    $G^{\prime }\leftarrow$GREEDY AREA(*S*)34:    **if** $w\left(G\right)\geq w\left(G^{\prime }\right)$ **then**35:        $S^{\prime }=G$36:    **else**37:        $S^{\prime }=G^{\prime }$38:    **end if**39:    **return** $S^{\prime }$40:**end function**

Algorithm 2 corresponds to the Greedy Budgeted Maximum Coverage algorithm for GIA. The input of GBMC corresponds to the collection of disks $S_{i}$ with radius *r* located at the center of every sensor $u_{i}$. In addition, $w_{i}$ is the associate weight $w_{i}$ of $S_{i}$ or the number of samples covered by $S_{i}$ including $u_{i}$. The output corresponds to the subset $S^{\prime }\subseteq S$ that maximizes the coverage within the budget *L*. The algorithm has two sub-routines: greedy area price in Lines 3–16, and greedy area in Lines 17–30. Greedy area price initializes the collection in *G* to empty in Line 4 and sets the initial spending value *C* to zero in Line 5. Line 6 sets the universal set to input collection *S*. Lines 7–13 present the optimization criteria to select a winner, namely by buying the sample $u_{i}$ with sample price $c_{i}$ that maximizes $\frac{W_{i}^{\prime }}{c_{i}}$ where $W_{i}^{\prime }$ is the number of samples within $S_{i}$ including $u_{i}$ but not covered by any set in G. In other words, select the sample price with the best coverage per dollar. Greedy area price, the second sub-routine, follows the same logic as the first one; it only differs in the optimization criteria to select the winner. Here, we buy the sample $u_{j}$ whose $S_{j}$ maximizes $W_{j}^{\prime }$, namely, maximizes the number of samples within $S_{i}$ including $u_{i}$ but not covered by any set in G. The GBMC runs the two sub-routines in parallel and selects the output that maximizes the number of samples covered. Finally, the GIA corresponds to a modified version of RADP-VPC-RC (Algorithm 1) where the criteria for winner selection (Lines 9–10) is substituted by the output of the GBMC (Algorithm 2).

#### 3.3.2. An Example of How GBMC for GIA Works

The following example shows how GBMC works and how it compares to the well-known Greedy Set Cover (GSC). Consider a scenario where *N* users are deployed in a target area with a given radius *r*. Let us take three users (sensors), $u_{1}$, $u_{2}$, and $u_{3}$, covering 1, 5, and 7 users, respectively, including themselves. In other words, their samples have weights $W_{1}^{\prime }=1$, $W_{2}^{\prime }=5$, and $W_{3}^{\prime }=7$. Assign the costs $c_{1}=1$, $c_{2}=3$, and $c_{3}=6$ for these samples, and define the total budget L=6.

#### Greedy Set Cover Algorithm Solution

The Greedy Set Cover (GSC) algorithm tries to maximize $\frac{W_{i}^{\prime }}{c_{i}}$ for i=1,2,3:$$\begin{array}{cc} \frac{W_{1}^{\prime }}{c_{1}} & =\frac{1}{1}=1,\\ \frac{W_{2}^{\prime }}{c_{2}} & =\frac{5}{3}\approx 1.67,\\ \frac{W_{3}^{\prime }}{c_{3}} & =\frac{7}{6}\approx 1.17. \end{array}$$

The GSC algorithm would first purchase $u_{2}$ because $\frac{W_{2}^{\prime }}{c_{2}}$ is the highest. However, after purchasing $u_{2}$, the remaining budget is 6−3=3, which is not enough to purchase $u_{3}$.

#### GBMC Algorithm Solution

In contrast, the GBMC algorithm evaluates two sub-routines (Lines 32–33):GREEDY AREA PRICE: Similar to GSC, it would select $u_{2}$, covering 5 users.GREEDY AREA: It selects $u_{3}$ because $W_{3}^{\prime }$ is the highest without violating the budget. It covers 7 users.

GBMC then compares the results of these sub-routines and ultimately chooses $u_{3}$, as it covers more users within the budget. Thus, the final decision by GBMC is to purchase $u_{3}$, which covers 7 users, maximizing the coverage better than the GSC algorithm under the given budget constraint.

#### 3.3.3. *k*-Greedy Algorithm

Algorithm 3 (k-greedy) enhances GIA by incorporating a brute-force component. Initially, it initializes two sets $H_{1}$, and $H_{2}$ (Lines 3–4). It sets $H_{1}$ with the singleton subset that maximizes the weight (Line 3), provided the cost is within the budget. Then, it starts $H_{2}$ by choosing a constant *k* and finds all *k*-size subsets of the participant group that can be afforded. For each subset, the GIA is executed, but it is forced to first select all participants in the *k*-subset. Finally, the algorithm compares the weights of $H_{1}$, and $H_{2}$ (Line 12), and outputs the *H* value with the higher weight. Thus, higher values of *k* significantly slow down the algorithm but yield results closer to the optimal. Section 3.3.4 provides a numerical example that details the step-by-step process of how k-greedy works.
**Algorithm 3** Improved *k*-greedy algorithm  1:**Input:** *S* a collection of sets, *L* budget, *k* an integer  2:**Output:** *H* a collection of sets  3:$H_{1}\leftarrow \arg\;\max\left\{w\left(G\right)|G\subseteq S,|G|<k,\;\mathrm{and}\;c\left(G\right)\leq L\right\}$  4:$H_{2}\leftarrow \emptyset$  5:**for all** G⊆S such that |G|=k and c(G)≤L **do**  6:    U←S∖G  7:    R←Greedy(U,L−c(G))  8:    **if** $w\left(R\cup G\right)>w\left(H_{2}\right)$ **then**  9:        $H_{2}\leftarrow R\cup G$10:    **end if**11:**end for**12:**if** $w\left(H_{1}\right)>w\left(H_{2}\right)$**then**13:    **return** $H_{1}$14:**else**15:    **return** $H_{2}$16:**end if**

#### 3.3.4. An Example of How *k*-Greedy Works

The following example illustrates the difference between GIA and greedy k = 2. Let $X=\{e_{1},e_{2},e_{3}\}$ be the set of samples associated with a set of mobile sensors (participants’ smartphones) and $S=\{S_{1},S_{2},S_{3}\}$ be the collection of sets, where each $S_{i}$ corresponds to the set of samples inside the disk centered at sensor $e_{i}$. Following our convention about the weight of sets (number of elements), and its cost (cost of sample $e_{i}$ at the center of the disk), we have
$S_{1}=\left\{e_{1},e_{2}\right\}$ with weight $w(S_{1})=2$ and cost $c(S_{1})=24$$S_{2}=\left\{e_{1},e_{2},e_{3}\right\}$ with weight $w(S_{2})=3$ and cost $c(S_{2})=42$$S_{3}=\left\{e_{2},e_{3}\right\}$ with weight $w(S_{3})=2$ and cost $c(S_{3})=24$

Let *L*=100 be the budget. Then, the total cost of the acquired samples should not exceed 100.

##### GIA Execution

Select the highest weight set: (Algorithm 2—line 34)
$S_{1}=\left\{e_{1},e_{2},e_{3}\right\}$ with weight $w(S_{2})=3$ and cost $c(S_{2})=2$.Current weight = 3.Remaining budget = 0.Remaining sets: $\left\{S_{1},S_{3},S_{4}\right\}$.No budget remaining to select another set:
Total weight = 3 (from $S_{2}$).

##### K=2-Greedy Execution

Calculate $H_{1}$: (Algorithm 3—line 3)
Single sets: $S_{1}$, $S_{2}$, $S_{3}$, $S_{4}$.Maximum weight from single sets = 3 from $S_{2}$.So, $H_{1}=\left\{S_{2}\right\}$ with weight 3.Calculate $H_{2}$: (Algorithm 3—lines 4–11)
Consider all subsets with cardinality exactly k=2:
$\left\{S_{1},S_{3}\right\}$ with weight 2+2=4 and total cost 1+1=2.$\left\{S_{1},S_{4}\right\}$ with weight 2+1=3 and total cost 1+1=2.$\left\{S_{3},S_{4}\right\}$ with weight 2+1=3 and total cost 1+1=2.Maximum weight from these combinations = 4 from $\left\{S_{1},S_{3}\right\}$.So, $H_{2}=\left\{S_{1},S_{3}\right\}$ with weight 4.Output the better solution between $H_{1}$ and $H_{2}$:
$H_{1}$ has weight 3, and $H_{2}$ has a weight of 4.Therefore, the output is $H_{2}=\left\{S_{1},S_{3}\right\}$ with weight 4.

Thus, while GIA ends up acquiring $S_{2}$ with a weight of 3, *k*-greedy acquires $\left\{S_{1},S_{3}\right\}$ with a weight of 4, maximizing weight and budget utilization.

#### 3.3.5. Pure Greedy

This approach uses the RADP-VPC-RC for data acquisition and the Pure Greedy Algorithm for winner selection. This simple strategy repeatedly selects the cheapest sample regardless of its location until the budget is exhausted. For the ease of the reader, we have sketched the Pure Greedy Algorithm. Here, the RADP-VPC-RC–Pure Greedy Algorithm corresponds again to a modified version of RADP-VPC-RC (Algorithm 1) where the criteria for winner selection (Lines 9–10) are substituted by the output of Algorithm 4.
**Algorithm 4** Pure Greedy Algorithm.  1:**Input:** *S* a collection of sets made up by the user locations  2:**Output:** $S^{\prime }\subseteq S$, covering set  3:**function** PURE GREEDY(*S*)  4:    G←∅  5:    C←0  6:    U←S  7:    **while** U≠∅ **do**  8:        select $S_{i}\in U$ with the lowest $c_{i}$  9:        **if** $C+c_{i}\leq L$ **then**10:             $G\leftarrow G\cup S_{i}$11:             $C\leftarrow C+c_{i}$12:        **end if**13:        $U\leftarrow U\setminus S_{i}$14:    **end while**15:    **return** *G*16:**end function**

#### 3.3.6. Random Algorithm

We also tested a random selection of participants. The Algorithm 5 repeatedly chooses randomly from a uniform distribution among the participants that it can afford until the budget is exhausted. The algorithm does not have any consideration in terms of location, price, or any optimization criteria.
**Algorithm 5** Random Selection Algorithm  1:**Input:** *S* a collection of sets made up by the user locations  2:**Output:** $S^{\prime }\subseteq S$, covering set  3:**function** RANDOM SELECTION(*S*)  4:    G←∅  5:    C←0  6:    U←S  7:    **while** U≠∅ **do**  8:        select $S_{i}\in U$ based on a random selection of $u_{i}$  9:        **if** $C+c_{i}\leq L$ **then**10:             $G\leftarrow G\cup S_{i}$11:             $C\leftarrow C+c_{i}$12:        **end if**13:        $U\leftarrow U\setminus S_{i}$14:    **end while**15:    **return** *G*16:**end function**

## 4. Performance Evaluation

This section outlines the performance evaluation for two sets of experiments. In the first set, the participants are pedestrians, whereas in the second set, the participants are represented as vehicles.

### 4.1. Pedestrian-Based Crowdsensing

For pedestrian-based crowdsensing, we evaluate two recruitment models: cheapest first (RADP-VPC-RC-pure-greedy) which corresponds to Algorithm 4, and cost-effective (maximum budgeted greedy-GIA) represented by Algorithm 2 for recruitment, and the recurrent reverse auction RADP-VPC-RC presented in Algorithm 1 as a data purchasing model.

#### 4.1.1. Experimental Setup

Table 1 outlines the parameters used in the experimental setup. The simulations take place within a square area of 100 × 100 units. The spatial distribution of participant locations follows three different scenarios: 2D normal distribution, uniform distribution, and exponential distribution. These distributions represent potential real-world situations. The 2D normal distribution for participant locations and bid prices simulates a stratified scenario, where a population is clustered in distinct neighborhoods (locations following a 2D normal distribution) with varying socioeconomic statuses (bid prices following a 1D normal distribution). In this case, participants in lower-income neighborhoods are expected to bid lower for their sensing samples compared to those in higher-income areas. Conversely, the uniform distribution for both participant locations and bid prices represents a scenario where neither location nor socioeconomic status has any influence on the participants’ asking prices per sample. Finally, the exponential distribution for locations and bid prices simulates a scenario where participants are densely clustered in certain regions, indicating higher population concentrations or activity levels. In such areas, increased competition may drive down bid prices, while participants in less populated or active regions might bid higher due to lower competition. The experiment employs a synchronous random model to handle participant mobility. Initially, participants are randomly deployed. For each round, the mobility algorithm updates their locations by calculating a displacement vector ($\vec{R}$), which combines the X-axis displacement ($\vec{X}$) and Y-axis displacement ($\vec{Y}$), such that $\vec{R}=\vec{X}+\vec{Y}$. The displacement vectors for each axis are generated using a random function (i.e., ${\left(-1\right)}^{\mathrm{uni}(0,2)}$), determining the movement direction. The speed is determined by the vector’s magnitude, which is derived by multiplying the number of steps (randomly selected between 0 and 3) by the step length (randomly selected between 0 and 11).

#### 4.1.2. Evaluated Metrics

We evaluate the following metrics: sensor radius, number of active participants, coverage, and percentage of budget utilization.

Sensor radius: We defined a sensor’s radius as the constant *r* that defines the area of coverage of each sensor. Thus, the acquisition of any sample $s_{j}$ in round *k* within the area coverage of sensor *i* is redundant if sample $s_{i}$ was already acquired in the same round.

Coverage: Coverage is defined not by the area, but by the number of participants (mobile sensors) within the area covered by the combined range of individual sensors associated with the acquired samples.

Number of Active Participants: This metric represents the average number of participants who continue to engage after *m* rounds of recurrent reverse auctions. Any crowdsensing system must maintain a minimum number of active participants to function effectively. A common issue with recurrent reverse auctions is the decline in participant numbers over time. This reduction leads to a spike in bid prices due to a lack of competition.

Percentage of Budget Utilization per Round: This metric assesses the average percentage of the budget used per round. A common issue is that a significant portion of the budget remains unused each round, leading to poor overall budget optimization.

#### 4.1.3. Experiments

This section describes a set of experiments conducted to test the performance of GIA optimizing a set of proposed metrics. The first set of experiments focuses on identifying the optimal radius length *r*. After establishing the ideal *r*, the next set of experiments involves comparing the average number of active participants and the average cost per round between the RADP-VPC-RC using a pure-greedy recruitment strategy and the GIA. Several distributions for users’ true valuations and participant’s locations were used. The final set of experiments aims to showcase the GIA’s superiority in terms of coverage, especially when participants are positioned in stratified locations within the area of interest.

#### 4.1.4. Experiment 1: Determining the Ideal Length of Sensor Radius *r*

To explore the relationship between radius *r*, the number of active participants, and the average price per round, *r* was incrementally increased from 1 to 10. Each scenario involved 1000 rounds, repeated 50 times, with results averaged under a fixed budget of 100 units per round. The average number of active participants was recorded for each value of *r* using three different distributions for users’ true valuations and a uniform initial deployment. The goal was to identify the radius at which participant numbers declined significantly. As shown in Figure 2 (right), when *r* exceeds 5, the number of active participants decreases consistently across all distributions. This occurs because a larger radius can encompass the entire population within one user’s sensing range, reducing the number of samples purchased and potentially leading to poor data representation. Conversely, a smaller radius may result in redundant data due to overlapping coverage. Thus, finding the right balance for *r* is crucial for optimizing data granularity and quality.

Figure 2 (left) shows that when *r* exceeds 5, the GIA fails to fully use the allocated budget per round. However, when r≤5, the algorithm continues using the entire budget. By combining the findings from these figures, we conclude that r=5 strikes a balance between maintaining a good level of participation and minimizing redundant data. In practice, however, the value of *r* should be adjusted based on how the variable of interest varies with distance.

#### 4.1.5. Experiment 2: Comparing the Performance Metrics

The purpose of this set of experiments is to quantify the number of active users within the system as the budget per round increases from 20 to 200 after 100 rounds. This analysis was conducted under three distinct user valuation distributions (sample bid prices), and participant distribution on the target area: a normal distribution with parameters μ=5 and σ=2, an exponential distribution with μ=5, and a uniform distribution within the range [0, 10]. The sensor radius was set to five meters.

Figure 3 (left), Figure 3 (center), and Figure 3 (right) illustrate that the GIA and RADP-VPC-RC-pure-greedy algorithm exhibit comparable performance, acquiring a similar number of samples per round across various budgets. These findings indicate that both algorithms effectively facilitate user participation and re-engagement within the system. However, despite the similar number of active users, the GIA adopts a distinct sampling strategy by avoiding the purchase of samples from proximate locations, thus preventing the acquisition of redundant data. Details of the experimental parameters are provided in Table 1.

#### 4.1.6. Experiment 3: Coverage, Number of Active Participants, and Cost

Here, we want to evaluate the performance of the GIA and RADP-VPC-RC-pure-greedy algorithms in terms of coverage, number of active participants, and cost versus the number of acquired samples. To achieve this, a stratified scenario is simulated, representing a situation where participants from varying socioeconomic statuses, each with distinct true valuations, reside in different regions of a city. In this context, the RADP-VPC-RC-pure-greedy algorithm tends to select samples that are not representative of the entire population and instead acquires redundant data. Conversely, the GIA successfully obtains non-redundant samples from each user and location type. The initial user deployment locations were generated using a 2D normal distribution, forming four clusters with means $\mu_{1}=\left(30,80\right)$, $\mu_{2}=\left(80,80\right)$, $\mu_{3}=\left(50,50\right)$, and $\mu_{4}=\left(90,30\right)$, and a common covariance matrix $\mathrm{Cov}=\left(\begin{array}{cc} 40 & 0\\ 0 & 40 \end{array}\right)$. Each cluster comprised 25 locations, totaling 100 locations. Users’ true valuations were generated from normal distributions with $\mu_{1}=5$, $\mu_{2}=10$, $\mu_{3}=15$, $\mu_{4}=20$, and σ=2. Each valuation cluster was associated with a corresponding location cluster to simulate the expected scenario.

Figure 4 (left) illustrates the percentage of area covered, Figure 4 (center) shows the number of active participants, and Figure 4 (right) displays the total cost per round as the number of acquired samples ranges from 5 to 50 in increments of 5. From Figure 4 (center) and Figure 4 (right), it is evident that both algorithms maintain a similar number of active participants and incur comparable expenditures to acquire their samples. However, Figure 4 demonstrates that the GIA algorithm selects samples from more widely distributed users across the area of interest, thereby achieving better coverage and minimizing redundant data acquisition. For instance, Figure 4 (left) indicates that when fifty samples are acquired, the GIA algorithm covers up to 64% more area than RADP-VPC-RC-pure-greedy.

### 4.2. Performance Evaluation for Vehicular Crowdsensing

In this section, we assess the effectiveness of four different recruitment or optimization approaches: Pure Greedy (Section 3.3.5): we will refer to this algorithm as RADP-VPC-RC because its original recruitment model is pure-greedy; cost-effective or GIA (Section 3.3.1): we will refer to this algorithm as GIA, k-greedy (Section 3.3.3), and Random Selection (Section 3.3.6). We use the RADP-VPC-RC as our purchasing model (Section 3.2) and the sensor model described in Section 3.1.

We evaluate the following metrics: number of active participants (i.e., users who remain engaged after 100 rounds of the recurrent reverse auction), coverage, and average total expenditure. The number of active participants reflects the sustained engagement and interest of users over time. Coverage assesses how effectively the auction mechanism captures a representative set of samples of the target areas of interest. Average total expenditure offers insights into the financial implications of the auction process. By analyzing these metrics, we aim to gain a better understanding of their performance and impact within the context of vehicular crowdsensing.

#### Experimental Setup

The elements of our simulation environment are illustrated in Figure 5. This environment comprises a map, a vehicular traffic simulator, and data on streets’ velocity distributions. We selected a densely mapped subsection of London’s drivable streets, enabling participants to take alternate routes and reach their destinations through different trajectories. The mapping is based on OpenStreetMap (OSM) [29], while the traffic simulation is conducted using SUMO [30]. SUMO provides realistic vehicle movement and routing algorithms applied to OSM data. Before importing a map into SUMO, we preprocess it using a discretization approach to correct any inconsistencies, following the method described by Goss et al. [31,32]. The parameters for this set of experiments can be found in Table 2. Additionally, we utilize the Uber Movement dataset [33] to estimate the velocity distribution of streets, which allows us to calculate average commuting times between any two locations on the map [34].

In these experiments, vehicles continuously traverse the map, joining and leaving the participation pool as they evaluate their Return on Investment (ROI) after each round. For each participant, we selected two points (a source and a destination) on a road map of Cologne, Germany, and had them drive between these two locations in a loop. In all experiments, the participants were split equally between the various districts, which corresponded to the centers of the multivariate exponential or normal distribution clusters used to place the participants. Unlike the previous pedestrian-based crowdsensing experiments where participant locations were drawn from a random distribution, it is not possible to sample two points from the distribution in this case, as most points will not fall directly on a road. Instead, we find the midpoint of each road, then evaluate the probability density function at that point and use that number as a weight for the road. From there, the probability that a road will be sampled is its weight divided by the sum of all weights. The covariance matrix used for the 2D normal distribution is $\left[\begin{array}{cc} 150 & 0\\ 0 & 150 \end{array}\right]$ For the exponential distribution, each road midpoint’s distance from the center of the cluster was calculated, and then the weight was calculated by evaluating the probability density function of an exponential distribution. The density function used was $\frac{\left(e-d\right)}{10}$ where d is the distance.

We conducted experiments to test the performance of our four recruitment approaches under five different scenarios. These scenarios corresponded to various combinations of participant trajectory locations and participant true valuation (participant sample expected value) distributions. In one scenario, we assumed that participants’ true valuation is heavily influenced by their location (location of their vehicles’ trajectories). This scenario may correspond to a city which divided into socioeconomically stratified neighborhoods or districts with different income levels. We modeled this by deploying vehicle trajectories following a 2D normal distribution (clusters) and generating participant true valuations also following a normal distribution. In a different scenario, we assumed participants’ true valuations were not influenced by the locations of vehicle trajectories. We simulated this by using a 2D normal distribution and a uniform distribution for the deployment of vehicle trajectories while generating participants’ true valuations from a uniform distribution. The first case may represent a city with an equitable income distribution, where regular vehicle trajectories are not correlated with participants’ true valuation. In the second sub-scenario, while participant locations remain unimportant, society may be stratified into different socioeconomic levels. This reflects a scenario in which city planners distribute public or subsidized housing units evenly throughout the city. Finally, we represent a scenario in which traffic density may influence the participants’ true valuations. The idea is that in areas with heavy traffic, the competition among participants’ vehicles may drive sample prices down, while the opposite may happen in areas with less vehicular activity. We simulate this scenario by using an exponential distribution for both the deployment of vehicle trajectories and the generation of participants’ true valuations. The implementation for all experiments can be found in https://github.com/gerzytet/BudgetedCoverage (accessed on 3 November 2024).

### 4.3. Experiment 2—Vehicles Trajectory Locations and Density Influencing Samples Bid Price

This experiment assesses the impact of budget allocation on coverage, the active number of participants, and the efficiency of budget utilization after 100 rounds of RADP-VPC-RC. We tested four recruitment approaches: pure greedy (selecting the cheapest options first), cost-effective (GIA), K-greedy (K = 1), and random selection. The experiment tested every budget level from 100 to 3000, in increments of 100, conducting 20 trials for each budget level. Figure 6 shows the results of evaluating these three metrics when both vehicle trajectory deployment and participants’ true valuations follow normal distributions.

On the other hand, Figure 7 shows the performance of our three metrics when vehicle trajectory deployment and participant bid prices follow exponential distributions.

Figure 6 shows that the cost-effective or GIA approach performs better than other recruitment approaches for budget values greater than 300. However, the GIA approach is closely followed by k-greedy. This result found on vehicle simulations is consistence with the one found with pedestrians, as shown in Figure 4 (left). In terms of the average number of participants after 100 rounds, the RADP-VPC-RC-pure-greedy is slightly better than GIA, which is not very different from the result with pedestrians. Finally, random acquisition seems to be the most efficient approach in terms of efficient budget expenditure. Figure 7 shows the simulation results of our three metrics under exponential distribution of vehicle trajectories, and participant true valuations. The result shows a similar trend in terms of the average number of active participants after 100 rounds, and the efficiency of budged utilization. However, there are small differences in terms of coverage where GIA outperforms k-greedy for a budget under 1700, and then k-greedy surpasses GIA for budget values greater than 1700.

In conclusion, for these two distributions, selecting a vehicle *i* may result in other vehicles falling within the area of influence (or coverage) of $u_{i}$. This performance can be attributed to the cost-efficient and *k*-greedy algorithms’ ability to optimize selection based on both cost and potential coverage, making it particularly effective in scenarios with clustered distributions of participants.

### 4.4. Experiment 2—Effect of Budget Allocation on Coverage, Active Number of Participants, and Budget
Utilization When Bid Prices Are Not Influenced by Trajectory Locations

The purpose of this experiment is to gain insight into how budget allocation affects coverage, the average number of active participants, and efficiency in budget utilization. All of them follow the assumption of no correlation between vehicle trajectory location and density, and sample bid prices. Here, we test three scenarios. In the first scenario, vehicle trajectories are distributed evenly through the city, and participant true valuations are uniformly distributed. The second scenario corresponds to vehicle trajectories that are uniformly distributed, but participant true valuations are normally distributed. Finally, the third scenario corresponds to vehicle trajectories that are normally distributed, namely people leaving and moving in well-defined districts while their true valuation is uniformly or evenly distributed.

Figure 8 shows the effect of budget allocation on coverage, average number of active participants, and efficient budget utilization, Here, we assume that vehicle trajectories and participant true valuations follow uniform distributions. Figure 9 presents the same comparison but this time, vehicular trajectories are uniformly distributed under the target area while the participant’s true valuation is normally distributed. Finally, Figure 10 corresponds to the case when the vehicle trajectories are grouped following a normal distribution while participant true valuations are uniformly distributed.

In the scenario where there is no apparent correlation between vehicle trajectory locations and participant true valuations, a few key patterns emerge. First, in terms of coverage, the k-greedy, GIA, and RADP-VPC-RC-pure-greedy approaches appear to perform at a similar level. Second, the RADP-VPC-RC-pure-greedy approach outperforms the other methods in maintaining a high number of active participants, which is consistent with the findings from the previous experiment. Finally, continuing the trend observed in prior experiments, the random acquisition approach seems to be the most efficient in terms of budget utilization.

## 5. Conclusions and Future Work

This paper introduces an incentive mechanism for vehicular crowdsensing based on greedy algorithms and a recurrent-based auction. By integrating dynamic incentive mechanisms, leveraging vehicular mobility patterns, and addressing budgetary constraints, our approach enhances data collection efficiency by reducing redundancy and ensuring broad spatial coverage, a high number of active participants, and efficient budget utilization. We show that greedy algorithms, while effective in pedestrian-based crowdsensing, require significant adaptation to perform well in the vehicular context. Using simulations grounded in realistic urban environments, benchmarking four recruitment strategies under different vehicular crowd distribution scenarios, and using extensive simulations, our methodology outperformed traditional methods by optimizing participant selection based on both geographical and bid sample prices. Going forward, future work will focus on refining these techniques further, particularly in dynamic environments with fluctuating participant availability and evolving coverage demands. In addition, we would like to perform experimentation in terms of the optimal sensor radius in the vehicular context. By continually improving our participant selection strategies, we aim to unlock the full potential of edge-assisted vehicular crowdsensing and deliver enhanced data collection capabilities.

## Figures and Tables

**Figure 1 sensors-24-07191-f001:**
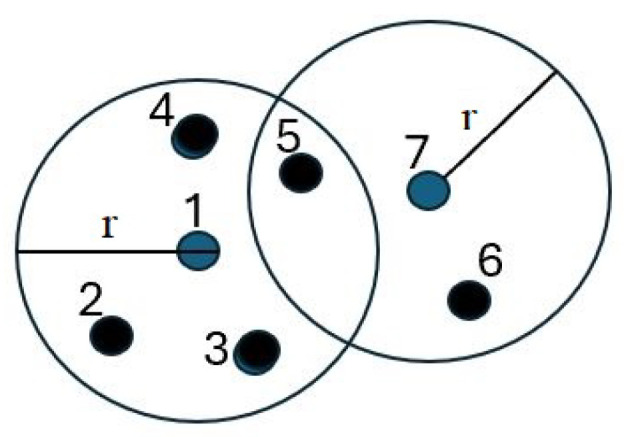
Example of coverage per user.

**Figure 2 sensors-24-07191-f002:**
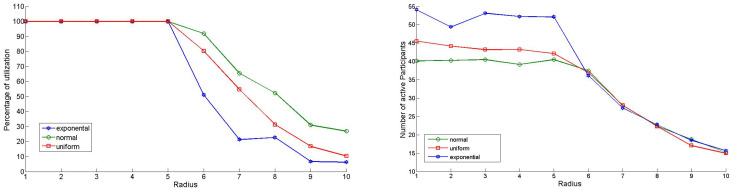
Radius vs. percent utilization (**left**) and number of participants (**right**).

**Figure 3 sensors-24-07191-f003:**
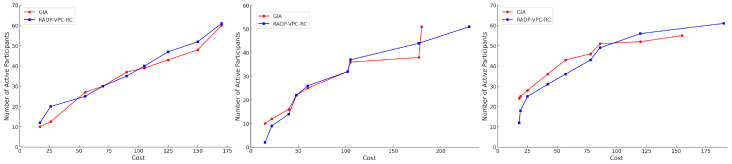
Cost vs. number of active participants under normal (**left**), exponential (**center**), and uniform (**right**) distributions.

**Figure 4 sensors-24-07191-f004:**
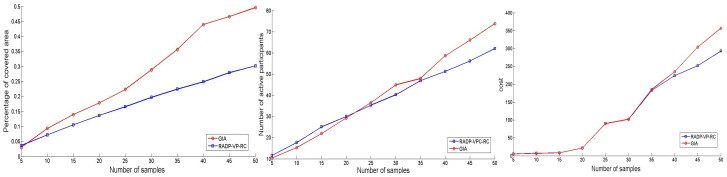
Number of samples vs. percentage area coverage (**left**), number of active participants (**center**), and cost (**right**).

**Figure 5 sensors-24-07191-f005:**
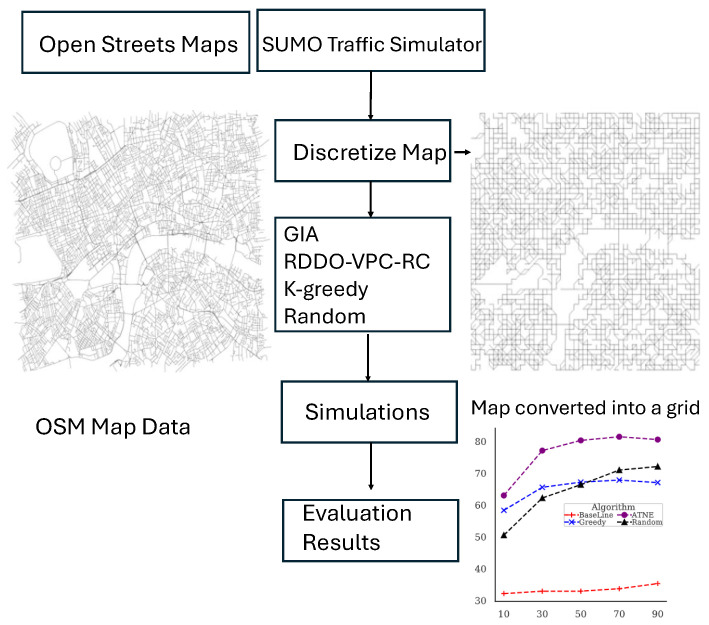
Simulation components.

**Figure 6 sensors-24-07191-f006:**
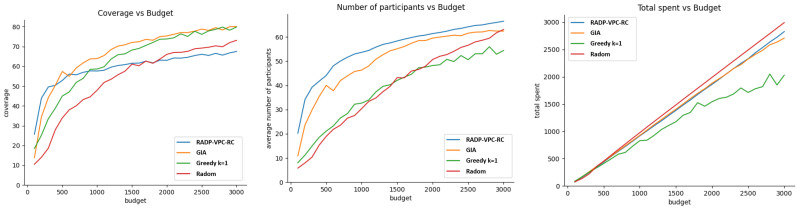
Normal distribution for trajectory distribution and participants’ true valuations.

**Figure 7 sensors-24-07191-f007:**
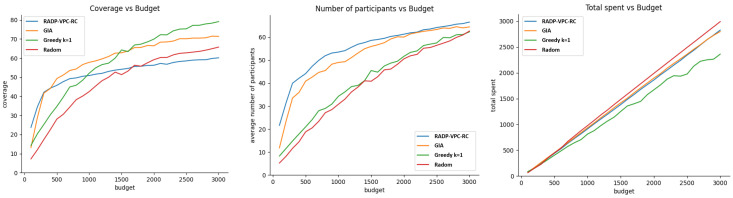
Exponential distribution for trajectory distribution and participants’ true valuations.

**Figure 8 sensors-24-07191-f008:**
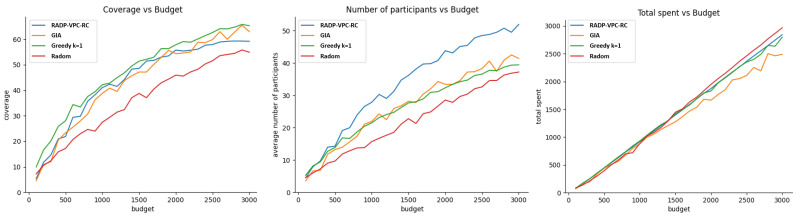
Budget vs. coverage, number of participants, and budget utilization under uniform distribution for trajectory locations and participant true valuations.

**Figure 9 sensors-24-07191-f009:**
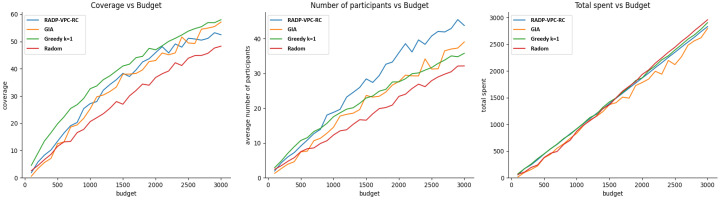
Budget vs. coverage, number of participants, and budget utilization under uniform and normal distributions for trajectory locations and participant true valuations, respectively.

**Figure 10 sensors-24-07191-f010:**
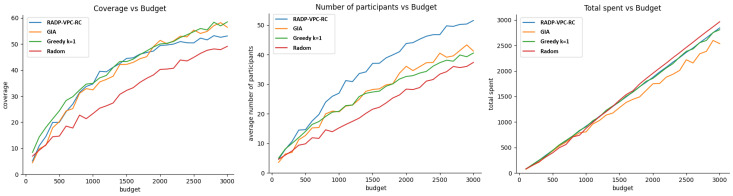
Budget vs. coverage, number of participants, and budget utilization under normal and uniform distributions for trajectory locations and participant true valuations, respectively.

**Table 1 sensors-24-07191-t001:** Parameters for simulation set 1.

Parameters	Experiment 1	Experiment 2	Experiment 3
Deployment area	100 m × 100 m		
Instances	100		
Deployment distribution	Uniform		
Deployment distribution			Four normal distributions with parameters:
			$\mu_{1}=\left(30,80\right)$, $\mu_{2}=\left(80,80\right)$, $\mu_{3}=\left(50,50\right)$, $\mu_{4}=\left(90,30\right)$, $Cov=\left(\begin{array}{cc} 40 & 0\\ 0 & 40 \end{array}\right)$
Uniform true valuation distribution	[0, 10]		No
Normal true valuation distribution	μ=5, σ=2		Yes
Exponential true valuation distribution	μ=5		No
Normal true valuation distribution			$\mu_{1}=5$, $\mu_{2}=10$, $\mu_{3}=15$, $\mu_{4}=20$, σ=2
RADP-VPC-RC	No	Yes	Yes
GIA	Yes	Yes	Yes
Radius *R*	1:10	5	5
Budget per round	100	20:200	0:350
Beta	(3, 7)	(3, 7)	(3, 7)
Alpha	Not used	7	Not used
ROI Threshold	0.5	0.5	0.5

**Table 2 sensors-24-07191-t002:** Vehicular-based crowdsensing simulation parameters.

Parameters		Parameters (Cont.)	
Target Area	5200 × 5200 m	Area Coordinates	
Cell Size	100 × 100 m	*Low Lat*	51.480861
		*Low Lon*	0.176478
Reward Distribution	normal, uniform, exp distributions	*High Lat*	51.539079
		*High Lon*	−0.057695
Vehicles trajectory deployment			
*Amount*	100		
*Source*	2D normal, uniform, exp distributions		
*Destination*	2D normal, uniform, exp distributions		
Algorithms	*Budget for ExperimentS 1–5*	*Experiment 50*	
*RADP-VPC-RC-pure-greedy*	range (start = 100, end = 3000, step = 500) ⋯		—
*GIA*	range (start = 100, end = 3000, step = 500) ⋯		—
*k-greedy, k = 1*	range (start = 100, end = 3000, step = 500) ⋯		—
*random*	range (start = 100, end = 3000, step = 500) ⋯		—

## Data Availability

https://github.com/gerzytet/BudgetedCoverage, accessed on 4 November 2024.

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
