# Peer review of "Bridging the Gap: An Algorithmic Framework for Vehicular Crowdsensing"

_sensors, 2024, doi:10.3390/s24227191_

Round 1

Reviewer 1 Report

Comments and Suggestions for Authors

The authors propose adapting existing incentive mechanisms, which have been previously in pedestrian crowdsensing, to vehicular crowdsensing. It is my opinion that the idea is interesting, and the topic would be of interest to readers of Sensors. However, I have some comments that I think should be addressed before the paper is ready for publication:

Suggestions for the authors, with no specific order:

- In lines 85-86, the authors state: "This study serves as a reference for establishing the parameters of the reverse auction scheme proposed here." The cited work (Danezis 2005) is almost 20 years old. I found surprising that a 20-year-old paper is used as the starting point for the parameters used during the experimentation. User's opinion and the importance given to privacy may have evolved significantly since then, are the more recent works that may serve as a reference?
- In table 1, why not explain the variables in later sections when actually using them?
Abbreviations can be explained when first referred using their meaning, as done later in the paper, example: Internet of Things (IoT)
- Broken references: I have seen 3 broken section references (lines 60, 310, 311). Please, carefully review all references in the paper and fix them as appropriate.
- Table 1: "Users or particpants" participants?
- Figure 1: r (lowercase) has been defined as the radius in line 169. However, in the figure, I think R represents the radius. I would put both either in lowercase or upper case for consistency.
- Figure 2: In X-Axis, "Radius" on left, "Radio" on right.
- Figure 3, left: Why does the GIA line go backwards at the end?
- 309-312: Section 4.1, I do not understand which algorithms are cheapest first (pure-greedy) and cost-effective (Maximum budgeted greedy). Are they named differently before?
- Lines 408-410: Where do all this values come from? Any reason for choosing μ1 = (30, 80), μ2 = (80, 80), μ3 = (50, 50), and μ4 = (90, 30)? Why do all clusters have the same since, 25 locations? It may be interesting to provide different datasets, ranging the number of locations, and seeing how different algorithms scale.  
- Something I have not seen mentioned in the experimentation section is the execution times of the different algorithms. How does the execution time or computational resources vary between RADP-VPC-RC and GIA?
- Figure 7, 8, 9: Why sometimes when the budget increases, the number of participants or the coverage decreases consistently? Wouldn't any solution valid for a given budget L be also valid for budget L + x, with x > 0?
- Line 541: Section title all in lowercase.
- Line 555: "senor", maybe sensors?
- Please consider publishing your artifacts following the FAIR principles. Specifically, publish the processed maps used as an input in your experiments, to facilitate further research in the topic. All instance data, and the executable / scripts used for solving the problem, including those used for analyzing the results and chart generation, should be published in repositories such as Zenodo. Optionally, consider publishing the full source code under an open license in a code repository, such as GitHub.

Comments on the Quality of English Language

There are several typos and inconsistencies in the manuscript. Carefully verify that Tables, Figures, and their corresponding text descriptions match both in name and nomenclature. Fix the broken section references.

Author Response

Reviewer#1, Concern # 1: In lines 85-86, the authors state: "This study serves as a reference for establishing the parameters of the reverse auction scheme proposed here." The cited work (Danezis 2005) is almost 20 years old. I found surprising that a 20-year-old paper is used as the starting point for the parameters used during the experimentation. User's opinion and the importance given to privacy may have evolved significantly since then, are the more recent works that may serve as a reference?

Author response:  Thanks for the important comment. The work of Danenzis is a reference in terms of how willing participants are to disclose their location.  However, the reviewer has a point in terms of including more recent studies.  We added a more up-to-date reference:   Schmitt, V.; Li, Z.; Poikela, M.; Spang, R.P.; Möller, S. What is your location privacy worth? Monetary valuation of different location types and privacy influencing factors. In Proceedings of the Proceedings of the 16th ACM Conference on Security and Privacy in Wireless and Mobile  Networks, 2023, pp. 19–29.

Reviewer#1, Concern # 2: In table 1, why not explain the variables in later sections when actually using them? Abbreviations can be explained when first referred using their meaning, as done later in the paper, example: Internet of Things (IoT)

Author response:  We have followed this approach throughout the paper, ensuring that all abbreviations are defined when they are first introduced, as in the example provided (e.g., Internet of Things (IoT)) by the reviewer.  We deleted Table 1 and ensured that every acronym was defined beforehand, providing clear and consistent explanations for variables and abbreviations upon their first use to enhance clarity and coherence

Reviewer#1, Concern # 3: Broken references: I have seen 3 broken section references (lines 60, 310, 311). Please, carefully review all references in the paper and fix them as appropriate

Author response:  We appreciate the reviewer bringing this to our attention. We have carefully reviewed the entire paper and have fixed the broken references, including those on lines 60, 310, and 311. All references are now correct and properly linked.

Reviewer#1, Concern # 4: Table 1: "Users or particpants" participants?

Author response:  We corrected this typo.

Reviewer#1, Concern # 5: Figure 1: r (lowercase) has been defined as the radius in line 169. However, in the figure, I think R represents the radius. I would put both either in lowercase or upper case for consistency.

Author response:  Thanks for the comment.  To keep consistency we change R by r in Figure 1.

Reviewer#1 Concern #  6:  Figure 2: In X-Axis, "Radius" on left, "Radio" on right.

Author response:  The typo has been corrected.

Reviewer#1, Concern # 7: Figure 3, left: Why does the GIA line go backwards at the end?

Author response:  The experiment was re-run, and a new figure was generated.

Reviewer#1, Concern # 8: Section 4.1, I do not understand which algorithms are cheapest first (pure-greedy) and cost-effective (Maximum budgeted greedy). Are they named differently before?

Author response:  Thank you for the observation. To maintain consistency, the cheapest-first approach, which corresponds to the pure-greedy recruiting method is the core of the RADP-VPC-RC algorithm. Therefore, we have updated the paper to refer to this as RADP-VPC-RC-pure-greedy in the text, while keeping RADP-VPC-RC in the graphs. Conversely, the cost-effective (maximum-budgeted greedy) approach is central to the GIA algorithm, so we modified the paper to refer to the cost-effective algorithm as GIA.

Author action:

Reviewer# 1, Concern # 10: Lines 408-410: Where do all this values come from? Any reason for choosing μ1 = (30, 80), μ2 = (80, 80), μ3 = (50, 50), and μ4 = (90, 30)? Why do all clusters have the same since, 25 locations? It may be interesting to provide different datasets, ranging the number of locations, and seeing how different algorithms scale.  

Author response:  Thank you for the observation. The rationale behind these parameters is as follows. The experiments are conducted on a 100x100 grid representing a city area. We aim to simulate a population living in distinct neighborhoods (cluster locations) with four socioeconomic statuses (true valuation). For this reason, clusters are distributed across the grid (at the corners and center) to represent various neighborhood types. For instance, a high-income neighborhood's location is generated randomly from a normal distribution with a mean of (x=90, y=30) and a specific covariance. This neighborhood is assigned a true valuation, also generated from a normal distribution with a mean of 20 and a standard deviation of 2 (high income). These parameters were selected to represent four neighborhoods, each with a different socioeconomic status. Additionally, we consider scenarios where location and socioeconomic status do not influence participants' true valuation. In these cases, we use uniform distributions to generate both locations and participants' true valuations.  Finally, while the parameters of the distributions are fixed, participants' locations and true valuations are randomly generated.

Reviewer# 1, Concern # 11: Figures 7, 8, 9: Why sometimes when the budget increases, the number of participants or the coverage decreases consistently? Wouldn't any solution valid for a given budget L be also valid for budget L + x, with x > 0?

Author response: Thank you for the valuable comment. While there are fluctuations, the overall trend shows that as the budget increases, both coverage and the number of active participants also rise. Several factors contribute to these fluctuations, the most significant being that we run 20 simulations for each budget level and average the results. Each simulation generates new vehicle source and destination trajectories, which can sometimes lead to lower coverage or fewer active participants. Coverage is heavily influenced by vehicle distribution: in one simulation, vehicles might be widely dispersed, so sampling may cover only a single vehicle; in another, several vehicles may be close together, allowing a single sample to cover multiple vehicles. In summary, these factors are largely driven by vehicle location and sensor radius. However, it is worth noting the consistent upward trend in coverage and the number of active participants with increasing budgets.

Reviewer# 1, Concern # 12: Line 541: Section title all in lowercase.

Author response:  We made all section titles in Pascal case and subsection titles in lowercase to be consistent.

Reviewer# 1, Concern # 13: Line 555: "senor", maybe sensors?

Author response:  Thanks for the comment. We corrected this typo.

Reviewer# 1, Concern # 14: Please consider publishing your artifacts following the FAIR principles. Specifically, publish the processed maps used as an input in your experiments, to facilitate further research in the topic. All instance data, and the executable / scripts used for solving the problem, including those used for analyzing the results and chart generation, should be published in repositories such as Zenodo. Optionally, consider publishing the full source code under an open license in a code repository, such as GitHub.

Author response:  We added the GitHub repository in line so the reader can reproduce the result presented in the paper.

Author action:

Reviewer# 1, Concern # 15: There are several typos and inconsistencies in the manuscript. Carefully verify that Tables, Figures, and their corresponding text descriptions match both in name and nomenclature. Fix the broken section references.

Author response:  Thanks for the valuable comment, we added the GitHub repository associated with this article in line 484

Reviewer 2 Report

Comments and Suggestions for Authors

The paper investigates whether greedy algorithms remain effective in the context of vehicular crowdsensing (VCS) as traditionally used for pedestrian-based crowdsensing. A dynamic incentive mechanism based on a recurrent reverse auction model, incorporating vehicular mobility patterns and realistic urban scenarios using the Simulation of Urban Mobility (SUMO) traffic simulator and OpenStreetMap (OSM) is employed to evaluate the applicability of successful participatory sensing approaches designed for pedestrian data and demonstrate their limitations when applied to VCS and show that greedy algorithms, while effective in pedestrian-based crowdsensing, require significant adaptation to perform well in the vehicular context. The study is interesting and is suggested to be accepted for publish after some mistakes are moved:

1. Page 2 line 60, “Finally, Section ?? summarizes” should be corrected.

2. Page 4 line 125, a dot should be added between “only one platform and multiple players Chakeri et al. [17,18]”.

3.Page 4 line 161 reviewer cannot find a mechanism in algorithm to guarantee the acquirement “SV S V ”.

Comments on the Quality of English Language

 Minor editing of English language required.

Author Response

Reviewer# 2, Concern # 1: Page 2 line 60, “Finally, Section ?? summarizes” should be corrected

Author response:  This broken reference to section 5 has been already fixed.

Reviewer# 2, Concern # 2: Page 4 line 125, a dot should be added between “only one platform and multiple players Chakeri et al. [17,18]”.

Author response:  Thanks for the important comment.  We correct this error, we introduce a dod to end the sentence “only one platform and multiple players. Chakeri et al. [17,18]...”

Reviewer# 2, Concern # 3: Page 4 line 161 reviewer cannot find a mechanism in algorithm to guarantee the acquirement “S∈V S ≈ V ”.

Author response:  Thanks for the valuable comment. We rephrased the sentence to avoid any ambiguity. The revised sentence reads: “Its goal is to acquire a sub-collection of samples, S = {S1, S2, . . . , SM}, that covers as many of the available samples as possible, resulting in a representative set of samples for the area of interest within a given budget.” For this purpose, we use a model (GIA) with three main components

Reviewer# 2, Concern # 4: Minor editing of English language required.

Author response: Thank you for the comment. We have thoroughly proofread the paper to correct any issues with grammar, structure, and style. 

Round 2

Reviewer 1 Report

Comments and Suggestions for Authors

I would like to thank the authors for their detailed response, I now think that the paper can be accepted for publication.